# The Corrosion Effect of Fly Ash from Biomass Combustion on Andalusite Refractory Materials

**Jozef Vlček** [1,2,*], **Hana Ovčačíková** [1], **Marek Velička** [1,2], **Michaela Topinková** [1], **Jiří Burda** [1,2] **and Petra Matějková** [3]

1 Department of Thermal Engineering, Faculty of Materials Science and Technology, VSB—Technical University of Ostrava, 17. listopadu 2172/15, 708 00 Ostrava, Czech Republic

2 Institute of Environmental Technology, CEET, VSB—Technical University of Ostrava, 17. listopadu 15/2172, 708 00 Ostrava, Czech Republic

3 Centre for Advanced Innovation Technology, VSB-Technical University of Ostrava, 17. listopadu 2172/15, 708 00 Ostrava, Czech Republic

* Correspondence: jozef.vlcek@vsb.cz; Tel.: +420-59732-1507

**Abstract:** The main problem affecting the life of refractory linings in furnaces is alkaline corrosion formed during biomass combustion, especially in systems with $SiO_2$–$Al_2O_3$. This corrosion effect is very intensive compared to using conventional technologies designed for burning traditional fuels. This study focuses on the development of a new type of andalusite refractory material with a higher corrosion resistance to $K_2CO_3$ and fly ash after biomass combustion. The original andalusite refractory material is labeled A60PT0, with an oxide content of 60 wt.% $Al_2O_3$ and 37 wt.% $SiO_2$, a compressive strength parameter of 64 MPa, and an apparent porosity of 15%. In the experiment, four mixtures (labeled A60PT1–A60PT4) were modified primarily using the raw materials and granulometry. The fly ash was characterized by an X-ray diffraction analysis with the following phases: quartz, calcite, microcline, leucite, portlandite, and hematite. According to the X-ray fluorescence analysis, the samples contained the following oxides: 47 wt.% CaO, 12 wt.% $K_2O$, 4.6 wt.% $SiO_2$, 3.5 wt.% MgO, and some minority oxides such as $P_2O_5$, MgO, MnO, and $Fe_2O_3$ between 2 and 5 %. The tendency for slagging/fouling of the ash was determined with the help of the indexes B/A, TA, $K_t$, and Fu. The final material was a shaped andalusite refractory material labeled A60PT4 with a content of 65 wt.% $Al_2O_3$ and 36 wt.% $SiO_2$. The properties of the andalusite material were a compressive strength of 106.9 MPa, an apparent porosity of 13%, and the recommended temperature of use up to 1300 °C. For corrosion testing, a static crucible test was performed according to the norm ČSN CEN/TS 15418 and the company's internal regulation. The exposure time of the samples was 2 h and 5 h at temperatures of 1100 °C and 1400 °C for $K_2CO_3$ and ash, respectively. For the evaluation of tested samples, an X-ray powder differential analysis, an X-ray fluorescence analysis, scanning electron microscopy, and energy-dispersive X-ray spectroscopy were used.

**Keywords:** refractory; fly ash; biomass; corrosion; andalusite

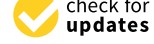



## 1. Introduction

Every year, the world generates 900–1000 million tons of energy waste [1] as coal, gas, oil, waste (municipal and industrial), and biomass. Today, biomass ranks among the fourth energy source in the world, where biomass accounts for about 14% [2]. According to the World Bioenergy Association [3], it makes up about 10% of the global energy supply and it is expected that by 2050, 33 to 50% of the world's energy reserves could be covered by burning biomass. A total of 476 mil tons of ash is formed from biomass combustion every year [4]. Depending on the type of combustion and the technology and fuel used, several solid residues are produced, which exhibit different properties. After high-temperature processing, solid residues have different chemical compositions, phase compositions, granulometries, volumes, and quantities, but also dangers. Several solid

residues are presented according to the authors of [5] in Table 1. The production of ash is, in these processes, unavoidable, and during combustion, changes occur to their physical and chemical properties, size, shape, etc. The result is a melting behavior [6] of original minerals and eutectics. During combustion, a solid ash phase is formed. Ash is a solid residue that leaves the boiler in the form of slag, cinder, or fly ash [7]. The fly ash fraction can be formed by coarse particles (particles larger than >1 μm) formed in a solid bed and fine particles (so-called particles <1 μm in diameter), which are primarily aerosols. Apart from solid residues, liquid-phase materials (liquid fuel droplets, tar, and water droplets) and gaseous-phase materials (CO, $H_2$, Cl, $SO_2$, $SO_3$, $N_2$, etc.) are formed [8].

**Table 1.** Chemical and phase composition of selected secondary energy waste (SEW) [5].

| Oxides/Type of Ashes | CaO | MgO | $SiO_2$ | $Al_2O_3$ | $Fe_2O_3$ | $K_2O$ | $Na_2O$ | $SO_3$ | $TiO_2$ | LOI |
|---|---|---|---|---|---|---|---|---|---|---|
| | wt.% | | | | | | | | | |
| A, GB, C | 5.5 | 1.1 | 33.6 | 17.8 | 13.2 | 3.6 | 0.23 | 1.02 | 1.43 | 22.2 |
| | quartz, mullite, lime, magnetite, periclase, $Fe_2MgO_4$ | | | | | | | | | |
| S, GB, C | 7.1 | 1.68 | 52.5 | 14.9 | 15.6 | 3.2 | 0.17 | 0.38 | 1.37 | 1.71 |
| | quartz, magnetite, hematite, anorthite ($CaAl_2Si_2O_8$) | | | | | | | | | |
| BA, FB, C | 39.6 | 0.67 | 20.8 | 10.5 | 7.2 | 1.63 | 0.12 | 15.9 | 1.22 | 1.5 |
| | anhydrite, quartz, lime, hematite, portlandite, gehlenite ($Ca_2Al_2SiO_7$) | | | | | | | | | |
| FA, FB, C | 37.9 | 0.6 | 20.4 | 10.9 | 8.3 | 1.57 | 0.14 | 6.89 | 1.42 | 10.9 |
| | quartz, lime, anhydrite, hematite, calcite, $CaAlO_4$ | | | | | | | | | |
| AGF, MW | 25.2 | 1.7 | 21.2 | 8.5 | 5.4 | 1.8 | 2.8 | 22.2 | * | 9.4 |
| | quartz, hematite, halite, sylvan (KCl), anhydrite, basanite ($CaSO_4 \cdot \frac{1}{2}H_2O$), graphite | | | | | | | | | |
| FA, FB, B | 37.9 | 7.4 | 16.3 | 2.5 | 2.5 | 12.25 | 7.05 | 2.81 | 0.21 | 4.2 |
| | quartz, calcite, lime, periclase, anhydrite | | | | | | | | | |

Notes: A—ash, S—slag, BA—bed ash, AGF—ash from grate fireplace, FA—fly ash, GB—granulation boiler, FK—fluid boiler, C—fuel coke, B—fuel biomass, MW—fuel municipal waste and *—not detected

The most common thermal disposal of biomass in the Czech Republic is incineration. Combustion is a physical–chemical process where heat is released, and the temperature of the burned materials increases. During this process, gases and other waste products are produced. The most known are reactions (Equations (1)–(3)) occurring during heterogeneous combustion. The oxidation of carbon occurs at higher temperatures. Carbon dioxide will then react with oxygen [9]. Heterogeneous combustion is a surface process that is affected by particle shape. Biomass particles have different shapes and sizes, and particles are not spherical. Non-spherical biomass particles have a larger surface area as well as significant porosity [10]. The models of biomass combustion are influenced by the shape of the particles and the surface of the particles during the combustion process [11].

$$C + O_2 \rightarrow CO_2 \ (-393 \ \Delta H \ kJ \cdot mol^{-1}) \tag{1}$$

$$CO + 1/2 O_2 \rightarrow CO_2 \ (-111 \ \Delta H \ kJ \cdot mol^{-1}) \tag{2}$$

$$H_2 + 1/2 O_2 \rightarrow H_2O \ (-242 \ \Delta H \ kJ \cdot mol^{-1}) \tag{3}$$

Biomass is one of the renewable energy sources (solid biofuels) that is an organic, non-fossil material of biological origin. For example, this includes wood waste, black liquor, bagasse, animal waste, and others [12]. In the Czech Republic, Wood as biomass is the most common combustion material. It accounts for about 64% of waste from the wood-processing industry, straw, cereals, and plant residues [13], with an average amount of ash as presented in Figure 1.

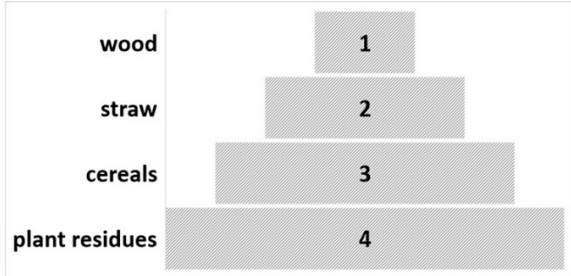
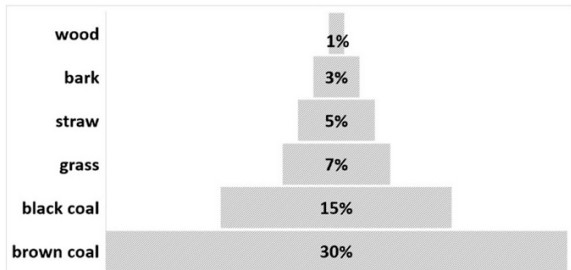

**Figure 1.** The ranking of biomass burning in the Czech Republic and the amount (%) of ash after combustion of different types of fuels.

In the Czech Republic, grate boilers are used for burning biomass. These furnaces are composed of both metal and refractory parts. The refractory materials are multi-component and heterogeneous ceramics with six oxides: $SiO_2$, $Al_2O$, $MgO$, $CaO$, $Cr_2O_3$, and $ZrO_2$, alone or in combination with carbon. Refractory materials are primarily used for protection against heat and mechanical and chemical corrosion [14]. Aluminosilicate materials are the most used for the application of linings. These can be divided into shaped or unshaped materials.

According to [15], alumina silica refractory materials is devided into groups according to the content of $Al_2O_3$ or $SiO_2$ (HA—high alumina, 45%–98% $Al_2O_3$; FC—fireclay, 30%–45% $Al_2O_3$; LF—low-alumina fireclay, 10%–30% $Al_2O_3$; SS—siliceous, 85%–93% $SiO_2$; and SL—silica, 93% $SiO_2$) [15,16]. The latter group can be subdivided into three subgroups according to the $Al_2O_3$ content: (1) mullite, 72–80 wt.% ($3Al_2O_3 \cdot 2SiO_2$), which is a solid solution with a molar ratio of $Al_2O_3$ to $SiO_2$ within the range of 3:2 to 2:1 [17]; (2) bauxite, 80–90 wt.% ($Al_2O_3 \cdot 2H_2O$); and (3) corundum ($Al_2O_3$), with contents >90 wt.% [18]. Mullite and corundum are the main refractory mineral grains [19]. High alumina has replaced better-quality firebricks in many applications [17]. Andalusite ($Al_2O_3 \cdot SiO_2$) is commonly used for preparing commercial refractory materials with high amounts of mullite. Andalusite and mullite are aluminosilicates with the general formula $Al_{4+2x}Si_{2-2x}O_{10-x}$. This group includes sillimanite, andalusite, and kyanite [20]. The heating of andalusite leads to the decomposition forming mullite and $SiO_2$ at 1100–1600 °C [20].

Andalusite materials can be described as follows [19]: the main oxide is $Al_2O_3$, comprising an amount of 60%–65%, which is combined with a low alkaline oxide; the main mineral phase in the $Al_2O_3$–$SiO_2$ system has various crystal structures and densities. The transformation into mullite and $SiO_2$ or the glassy phase starts at approx. 1250 °C, and this step depends on the chemical composition and the grain size. The volume is changed with sillimanite and andalusite (approx. 5%–8%), and unfired andalusite is used [19]. As presented by the authors of [21], andalusite has an excellent resistance to alkalis due to the mullite network and the presence of amorphous $SiO_2$ after the absorption of the alkaline vapor. It is known that an improvement in alkali resistance results in optimizing the structure of pores and the chemical composition [22].

The operating conditions in boilers can be different. Above all, the goal is to limit high combustion temperatures and NOx in the flue gas and to prevent ash from sticking to the chamber walls, thus eliminating corrosion and erosion. From a design point of view, the combustion chamber of the boiler should be adapted so that the flame temperature drops below the melting temperature of the ash [23]. Heat-resistant linings are in direct contact with the thermally processed material. During the heating of the lining, a wide temperature gradient is formed, where the surface temperature affects the rate of the corrosion reactions.

The corrosion mechanism is a complicated process, and it occurs in cases of (a) attack by melting, (b) the penetration of flue gases into the lining, (c) the adhesion of dust particles to the lining, or (d) an instability of temperatures in the combustion chamber. The many authors and application companies precisely in connection with aluminosilicate refractory materials have solved the problem of so-called "alkaline corrosion/alkali-bursting" [24–26].

The consequences of that negative phenomenon are shown in Figure 2, with photos from different combustion devices after biomass combustion. The high proportion of alkali metal compounds represents a serious problem [27]. Alkali compounds contained in biomass ash promote the formation of new phases in the A–S system, which is the reason for volume changes and mechanical damage to linings (roof, walls, etc.).

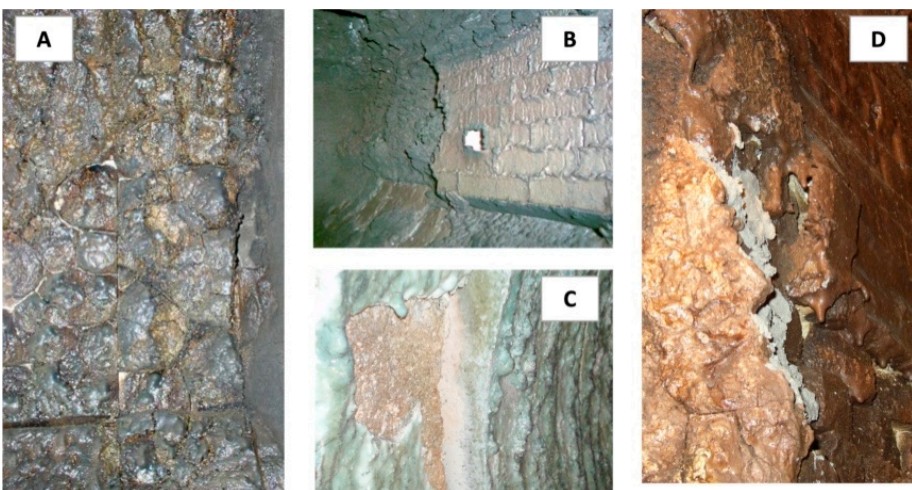

**Figure 2.** Degradation and corrosion of refractory materials in boilers after biomass combustion; (**A**) lining with corroded sticker after 1 year of biomass combustion; (**B**) corroded part of andalusite refractory samples after 2 years of combustion of plant biomass; (**C**) broken refractory concrete lining; and (**D**) boiler for uncorking—sticker and corroded walls [28,29].

Fireclay products contain less than 45% $Al_2O_3$, as opposed to high-alumina refractory products, which contain more than 45% $Al_2O_3$. High-alumina refractory materials containing mullite ($3Al_2O_3 \cdot 2SiO_2$ as $A_3S_2$) have a different process of reactions than fireclay materials with an $Al_2O_3$ content below 45%. Na-aluminosilicate phases are formed by a $Na_2O$ attack. In most cases, oxide $SiO_2$ is attacked first according to the reaction shown in Equation (4) in the A–S system [30]. If its free silica is consumed and its mullite ($Al_6Si_2O_{13}$) is attacked, albite is formed ($NaAlSi_3O_8$) by the reaction shown in Equation (5) [30,31]. Mullite is often present in fireclay materials together with cristobalite ($SiO_2$); it reacts with $NaO_2$ above 1000 °C, forms a nepheline phase ($NaS_2$), and forms $\alpha$-$Al_2O_3$ according to the reaction in Equation (6):

$$Na_2SO_4 + 2SiO_2 = Na_2Si_2O_5 + SO_2 + 1/2O_2 \tag{4}$$

$$Na_2Si_2O_5 + 2Al_6Si_2O_{13} = 2NaAlSi_3O_8 + 5Al_2O_3 \tag{5}$$

$$3Al_2O_3 \cdot 2SiO_2 + Na_2O \rightarrow Na_2O \cdot Al_2O_3 \cdot 2SiO_2 + 2Al_2O_3 \tag{6}$$

The formation of $\beta$-alumina after the reaction of $Na_2O$ with free $\alpha$-$Al_2O_3$ is presented by greater expansion and internal stress. The compound $Na_2O \cdot 11Al_2O_3$ is referred to in technical practice as $\beta$-alumina. From a mineralogical point of view, it is listed as diaoyudaoite [32], which forms above 1100 °C. The process is associated with an increase in the volume by up to 18%. The attack of alkali $K_2O$ in the aluminosilicate system leads to the formation of K-aluminosilicate phases. According to the authors of [33] and Equation (7), this is a present reaction between potassium vapor and a refractory material.

The authors of [34] describe the potassium attack in refractory materials with a high content of $Al_2O_3$, in which $\beta$-alumina and then potassium aluminate arise after the reaction in Equation (8). The reaction of Si with kaliophilite can then take place according to Equation (9). The reaction of potassium silicate with mullite is described by Equation (10), where continued volume changes and expansion of between 20 and 25% occur. Kaliophilite, $KAlSiO_4$, is formed within 30 min at a temperature of 1000 °C and causes flaking and the

expansion of linings. At an $Al_2O_3$ content >30%, a new phase of leucite $KAS_4$ ($KAS_4$) is formed according to Equation (11), which has a melting point of up to 1693 °C [32,35–38].

The formation of orthoclase $KAS_6$ ($KAS_6$) is more favorable at a lower content of $Al_2O_3$ (<30%), with an incongruent melting point at 1150 °C. Other products from biomass combustion include water vapor, sulfur oxide, and Cl. Cl significantly shortens the life of the lining, e.g., corrosion from $FeCl_2$ or $ZnCl_2$ [39]. A higher concentration of $SO_2$ in the flue gas causes alkali sulphation with decreasing flue gas temperatures.

$$6K + 3Al_2O_3 \cdot 2SiO_2 + 4SiO_2 + 3CO \rightarrow 3(K_2O \cdot Al_2O_3 \cdot 2SiO_2) + 3C \tag{7}$$

$$Al_2O_3 \rightarrow K_2O \cdot 12Al_2O_3 \rightarrow K_2O \cdot Al_2O_3 \tag{8}$$

$$K_2O \cdot Al_2O_3 \cdot 2SiO_2 + 2SiO_2 \rightarrow K_2O \cdot Al_2O_3 \cdot 4SiO_2 \tag{9}$$

$$3(K_2O \cdot 2SiO_2) + 3Al_2O_3 \cdot 2SiO_2 \rightarrow 3(K_2O \cdot Al_2O_3 \cdot 2SiO_2) + 2SiO_2 \tag{10}$$

$$K_2O \cdot Al_2O_3 \cdot 6SiO_2 \rightarrow K_2O \cdot Al_2O_3 \cdot 4SiO_2 + 2SiO_2 \tag{11}$$

## 2. Materials and Methods

### 2.1. Chemical and Phase Composition of Andalusite Refractory Material Labeled A60PT0

Andalusite refractory material has high compressive strength characteristics, a high load capacity in heat, and good dimensional stability. However, it has a lower density, i.e., a higher apparent porosity of 15% associated with lower corrosion resistance and easier penetration into the refractory material. At first, the original sample labeled A60PT0 was tested.

This sample belonged to the group of aluminosilicate refractory samples with a high content of $Al_2O_3$. The chemical composition is shown in Table 2. The parameters measured for this material were the bulk density, 2550 kg/m³; refractories under load (RUL) $T_{0.5}$, 1600 °C; cold compressive strength, CS = 64 MPa; and apparent porosity, 15%.

**Table 2.** Chemical composition and properties of andalusite A60PT0 refractory material (wt.%).

| Raw Materials | $Al_2O_3$ | $SiO_2$ | $P_2O_5$ | MgO | CaO | $K_2O$ | $Na_2O$ | $TiO_2$ | $Cr_2O_3$ | $Fe_2O_3$ |
|---|---|---|---|---|---|---|---|---|---|---|
| | 60.9 | 37.3 | 1.3 | 0.1 | 0.1 | 0.6 | 0.1 | 0.4 | 0.1 | 1.9 |

The andalusite refractory material contained 60 wt.% $Al_2O_3$ and 37 wt.% $SiO_2$. Generally, more impurities such as $Na_2O$, $K_2O$, CaO, MgO, and $Fe_2O_3$ relate to the raw andalusite and the geographical location [17]. The crystalline phases are recorded in Figure 3. The andalusite sample A60PT0 contained phases such as mullite, andalusite, corundum, and $SiO_2$ in form cristobalite. Andalusite and quartz phases were identified too.

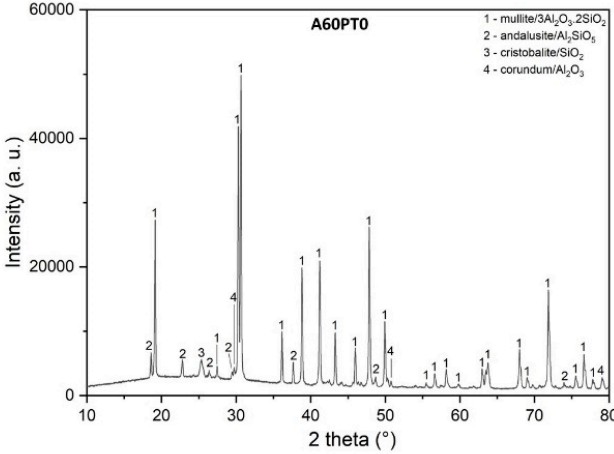

**Figure 3.** Phase composition of original andalusite refractory materials.

### 2.2. Characterization of Fly Ash from the Combustion of Woodchips

Fly ash was used in the experiment in a heterogenous mixture after biomass combustion as a corrosion agent, as shown in Figure 4. It was obtained from the Czech Republic. During the experiment, fly ash was used in its original form, without mechanical treatment or sieving, primarily for the crucible corrosion test. More chemical and phase information about the ash is present in Table 3 and Figure 4. Powdered fly ash has a variable chemical composition and can be considered an unrefined material. The main oxides were CaO, at 47%, and $K_2O$, at 12%. The minority oxides were $SiO_2$, $P_2O_5$, MgO, MnO, and $Fe_2O_3$, comprising between 2 and 5%. The amount of alkaline material present for corrosion attack/degradation was significant. In this case, the fly ash had a content of 12.2% $K_2O$ + $N_2O$.

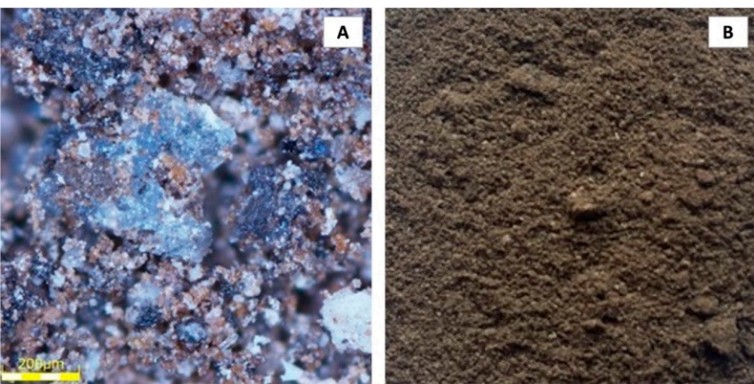

**Figure 4.** The original tested fly ash; (**A**) details of the fly ash under a microscope and (**B**) original ash after woodchip combustion.

**Table 3.** Chemical composition of tests of fly ash after the combustion of woodchips.

| Oxide wt.% | $Al_2O_3$ | $SiO_2$ | $P_2O_5$ | MgO | CaO | $K_2O$ | $Na_2O$ | $TiO_2$ | $Cr_2O_3$ | $Fe_2O_3$ | MnO | BaO | LOI |
|---|---|---|---|---|---|---|---|---|---|---|---|---|---|
| | 0.9 | 4.6 | 2.5 | 3.5 | 47.6 | 12.0 | 0.2 | 0.1 | 0.1 | 2.7 | 3.5 | 0.3 | 21 |

The fly ash from biomass combustion has, in many cases, low melting temperatures due to the high content of alkali oxides and Cl. There was no Cl detected in the sample. A corrosion attack by Cl reduces the service life of refractory linings. From the point of view of phase compositions, as presented in Figure 5, ash was very variable, with seven phases detected. One dominant peak with a high frequency was calcite, followed by microcline, portlandite, orthoclase, leucite, hematite, and quartz.

The analyzed fly ash had a relatively higher LOI (loss on ignition) value, which is caused by the content of the combustible substances or the under-burning of the original material. The size of this proportion of organic substances is mainly influenced by combustion technologies, types of boilers, etc. An LOI above 7 wt.% can also be caused by unregulated or excessively high drafts of flue gases through the chimney when a substantial part of the burnt sawdust or straw also leaves the ash separator. If the ash is examined after all combustible substances have been burned, then the LOI includes the breakdown of the crystalline structure of clays (removal of crystalline water), the breakdown of carbonates (limestone, dolomitic limestone, and dolomite), and any other substances [40].

In the case of ash, it is also possible to determine the fusibility of the ash according to the coefficient of the so-called Tuene number, which is the ratio of acidic and basic oxides. [7].

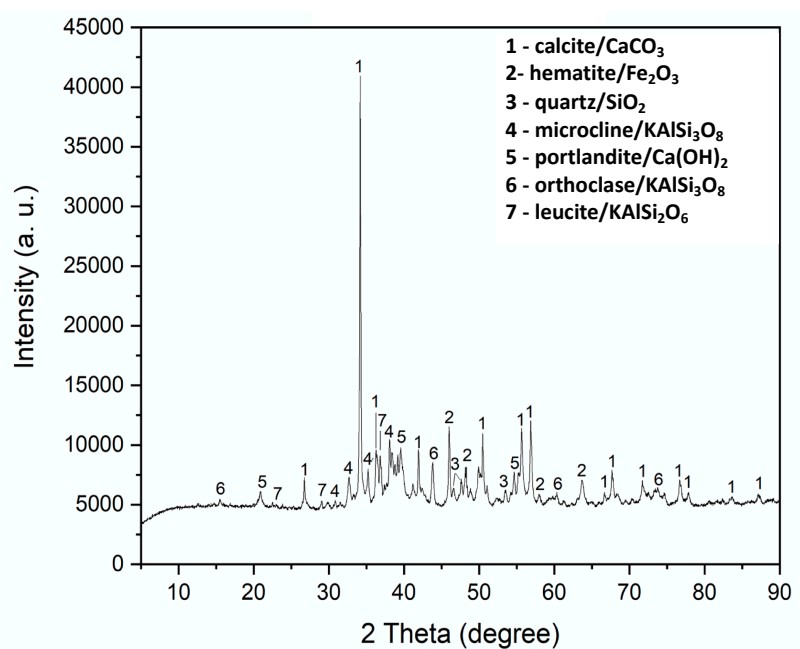

**Figure 5.** Phase composition of fly ash after woodchip combustion.

During biomass combustion, slag is formed, set, and deposited in combustion units. In this case, it is possible to use indexes for slagging and fouling. Indexes are calculated from the elementary chemical composition of the ash. Although these indexes are applied mainly for coal combustion, many authors also use them for biomass ash. There are the index of based-acid ratio (B/A), the index total alkali (TA), and, for example, the fouling index (Fu), which is calculated from a chemical composition according to the formula presented in Table 4 [41–43] of the analyzed materials.

**Table 4.** Characterization of fly ash based on Tuene number and indexes of slagging/fouling [41–43].

| Tuene Number | Index of Base–Acid Ratio | Total Alkalis | Fouling Index |
|---|---|---|---|
| $Kt = \frac{S+A}{F+C+M}$ | $B/A = \frac{F+A+C+M+N+K}{S+A+T}$ | $TA = N + K$ | $Fu = \frac{B}{A}(N+K)$ |

Note: S—$SiO_2$, A—$Al_2O_3$, F—$Fe_2O_3$, C—CaO, M—MgO, N—$Na_2O$, K—$K_2O$, and T—$TiO_2$.

A value of $K_t$ = 0.10 is defined as easily fusible ash because the value is less than 2.4, which corresponds to the definition of the $K_t$ index as "easily fusible". With a temperature between 1150 and 1400 °C, it was defined as a "medium" $K_t$, and a value of $K_t$ > 2.5 with an ash melting temperature of 1400 °C was defined as "hardly fusible" ash.

*Index B/A (the alkalinity index)* is based on the general rule that basic oxides lower the melting temperature of ash while acidic oxides increase their temperature. A B/A value = 11.7 is defined as "extremely high". A value of B/A < 0.5 is "low", indicating that the tendency for slagging/fouling of fly ash will be great. As a final result, fly ash is strongly alkaline due to its high CaO content.

Index *TA (the total alkali index)* assesses the formation and ash deposits. For the fly ash, the TA value was 12.2, which is a high value. In general, a value >0.4 is higher, and a value <0.3 is low. Index *Fu (the fouling index)* expresses the amount of alkali content. Alkalis form a eutectic in combination with $SiO_2$. High values (*Fu* > 40) correspond to higher fouling tendencies. This fly ash achieved a value of *Fu* = 23.9, which is a medium value.

### 2.3. Characterization of $K_2CO_3$

The second corrosive medium that was used was anhydrous potassium carbonate. This product was from the company Penta s.r.o., Czech Republic. Information regarding

the chemical and physical properties is declared in its safety data sheet: color—white, state—solid, melting point—891 °C, and content in wt.%—>99.

### 2.4. Characterization Method

The chemical composition (XRF) of fly ash was determined using the method of energy-dispersive X-ray fluorescence spectroscopy (ED-XRF) on a SPECTRO XEPOS (Spectro Analytical Instruments, Kleve, Germany). Powdered samples were shaped and pressed into tablets for the XRD measurement. The mineralogical composition (XRPD) of the samples was evaluated using an X-ray diffraction analysis on the X-ray diffractometer MiniFlex 600 (Rigaku, Japan) equipped with a Co tube and D/teX Ultra 250 detectors. The XRD patterns were recorded over the 5–90° 2θ range with a scanning rate of 5°×min. The morphology of particles was studied using the scanning electron microscope QUANTA 450 FEG (FEI, Hillsboro, OR, USA). All images were collected using a secondary electron detector. An accelerating voltage of 25 kV was used. Each sample was Au/Pd-sputtered before the analysis.

### 2.5. Corrosion Test of Andalusite Refractories and Evaluation

The corrosion resistance of andalusite materials was performed by a crucible corrosion test, as shown in Figure 6. $K_2CO_3$ and fly ash were used as the corrosion agents. The conditions and evaluation of the experiment proceeded according to the methodology by ČSN P CEN/TS 15418 [44] and the author of [45]. Uniform regulation for the corrosion testing of refractory materials does not exist. The shape and test temperatures can be changed depending on the customer's requirements, the experiment, or the company's internal regulations [45]. For the testing, a cube was prepared with the parameters of 5 × 5 × 5 cm and a hole in the middle with a diameter of 2 cm. The corrosion agents were $K_2CO_3$ and fly ash. In this study, the following corrosion regimes were applied for the testing of an andalusite refractory material:

1. Start up for 5 h/maximum temperature of 1100 °C/maintain for 5 h/slowly cool/ corrosion agent was 5 g of $K_2CO_3$ for samples A60PT0 and A60PT4.
2. Start up for 5 h/maximum temperature of 1100 °C/maintain for 2 h/slowly cool/ corrosion agent was 20 g of $K_2CO_3$/tested samples A60PT1, A60PT, and A60PT3.
3. Start up for 5 h/maximum temperature of 1400 °C/maintain for 5 h/slowly cool/ corrosion agent was 5 g of fly ash/tested samples A60PT0 and A60PT4.

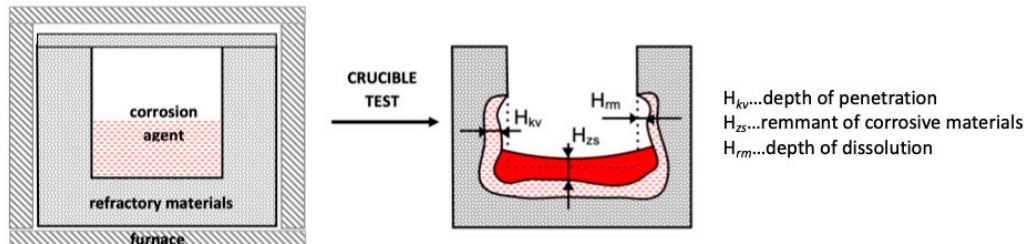

**Figure 6.** Crucible test of andalusite refractory material.

After cooling, the crucible was vertically cut and the penetration of refractory material was measured. The affected area was otherwise evaluated. The advantages of the test are its simplicity and quick results. The evaluation of corrosion tests can often be subjective, and often depends on a visual evaluation and operator or research experience. This study was evaluated according to a norm [44] with individual classes described in Table 5 and an internal evaluation in Table 6. Both regulations were implemented on the tested materials.

**Table 5.** Corrosion evaluation of refractory materials after ČSN P CEN/TS 15418 [44].

| Description of Corrosion Classification Test | Class |
|---|---|
| unaffected/no visible attack | U |
| lightly attacked/minor attack | LA |
| attacked/clearly attacked | A |
| corroded/completely corroded | C |

**Table 6.** Corrosion evaluation of refractory materials after internal regulation [45].

| | Corrosion Attack/Infiltration | Cracks | Class |
|---|---|---|---|
| | no changes | no | A |
| + | <6 mm corrosion and/or infiltration | no | B |
| ++ | >7 mm corrosion and/or infiltration | slight | C |
| +++ | >9 mm corrosion and/or infiltration | large, visible | D |

## 3. Results and Discussion

*3.1. Experiment I: Modification of Original Mixture of Andalusite Refractory Material A60PT0 and Preparing Four New Mixtures*

The expected modifications of the original andalusite material A60PT0 were adjustments to its properties. Sometimes, this material is used in a furnace where biomass is burned with no optimal or longtime effect. The aim was to increase the corrosion resistance, namely, to increase the amount of $Al_2O_3 \geq 60\%$, reduce the apparent porosity (AP) value to $\leq 12\%$, and be stable during the application at a temperature of 1300 °C. In Table 7, four recipes of modified mixtures are shown (labeled A60PT1–T4). The chemical compositions of these adjusted mixtures are also indicated in Table 8. The original sample for developing the four new compounds (labeled A60PT1–T4) was A60PT0.

**Table 7.** Raw composition of andalusite refractory mixtures A60PT1, A60PT2, A60PT3, and A60PT4.

| Mixtures | EC | BB | AN | RA | C | H$_3$PO$_4$ | P | AP | BD | CS |
|---|---|---|---|---|---|---|---|---|---|---|
| | (%) | | | | | | | (%) | (kg/m$^3$) | (MPa) |
| A60PT1 | - | - | x | x | x | x | - | 16.2 | 2518 | 61 |
| A60PT2 | x | x | x | - | x | x | - | 12.6 | 2612 | 70.6 |
| A60PT3 | x | x | x | - | x | x | - | 13.7 | 2656 | 87.5 |
| A60P74 | - | - | x | - | x | - | x | 11.7 | 2679 | 106.9 |

Note: EK—electro fused corundum, BB—burnt bauxite, AN—andalusite, RA—reactive Al$_2$O$_3$, C—clay, P—plasticizer, AP—apparent porosity, BD—bulk density, and CS—compressive strength.

**Table 8.** Chemical composition of andalusite refractory mixtures A60PT1, A60PT2, A60PT3, and A60PT4.

| Mixtures | SiO$_2$ | Al$_2$O$_3$ | TiO$_2$ | Fe$_2$O$_3$ | CaO | MgO | K$_2$O | Na$_2$O |
|---|---|---|---|---|---|---|---|---|
| | (%) | | | | | | | |
| A60PT1 | 34.36 | 63.23 | 0.29 | 0.81 | 0.1 | 0.16 | 0.48 | 0.20 |
| A60PT2 | 34.05 | 63.33 | 0.43 | 1.38 | 0.24 | 0.15 | 0.48 | 0,.1 |
| A60PT3 | 30.00 | 67.66 | 0.35 | 1.52 | 0.24 | 0.13 | 0.45 | 0.21 |
| A60PT4 | 32.65 | 65.06 | 0.34 | 1.38 | 0.21 | 0.14 | 0.42 | 0.25 |

The first step was to modify the granulometry to increase the density of the refractory materials. The new granulometries of the compositions were a combination of several fractions: the first change was the addition of sub-mesh fractions under 0.09 mm (~30%); the second change was an increase in the over-mesh with grains of 1 mm (~37%) and an increase in the coarse fraction of andalusite with grain sizes of 1–4 mm.

Furthermore, an amount of phosphoric acid was added to a volume of 10–50 mL of 28% $H_3PO_4$. Metal oxides react with phosphoric acid to form salt as a form of bonding in mixtures. $Al_2O_3$ and $H_3PO_3$ reacted too slowly at ordinary temperatures. Between the temperatures 127 °C and 427 °C, the process of phosphoric acid dehydration continued, and $AlPO_4$ was formed between the temperatures 732 °C and 1327 °C [46]. Firing above 1400 °C created a glass phase and influenced the higher strength of the samples.

The next step was the creation of a composition of refractory material, i.e., a high amount of aluminum represented by andalusite fused together with corundum and a ceramic matrix containing a lower content of $Al_2O_3$. It was assumed to have a higher resistance to alkaline corrosion due to the elimination of the formation of a new phase with a higher melting temperature than the application. The goal was to obtain a reasonable reaction of alkalis to form a melt, which would secondarily close the refractory material and prevent further corrosion of the refractory material.

Mixture A60PT1 was modified to the grain size curve described below, with the raw materials sillimanite and reactive $Al_2O_3$ as the input materials. This idea turned out to be unsuitable. The sample volume increased by 1.8% compared to the standard andalusite sample, A60PT0, which had a negative effect on increasing the apparent porosity to 16.2% compared to the current product. The lower compressive strength of the sample, only CS = 61 MPa (Table 8), was the result of the reaction of a higher proportion of clay bonds with reacting $Al_2O_3$ additives. The formed mullite phase did not allow sintering, i.e., compaction of the samples during the firing of the sample.

Mixture A60PT2 was created with a middle fraction of between 0 and 1 mm of electro-fused corundum, bauxite in the amount of 15 wt.%, and andalusite to provide resistance to melting and abrasion. However, this recipe did not show the expected parameters, even though the apparent porosity decreased by 3.6%. The parameters of compressive strength were over 70 MPa compared to a sample of mixture A60PT1. The mixture A60PT3 contained the same composition of raw materials as sample A60PT2, but the electro-fused corundum was increased to 10%. Although the compressive strength of sample AP0PT3 increased to CS = 87.5 MPa, the apparent porosity increased to 13.7% at the same time.The lower values of compressive strength of the sample A60PT1 and A60PT0, (Table 8), was apparently the result of a higher proportion of clay bonds reacting with $Al_2O_3$ additives.

This idea turned out to be unsuitable. The sample volume increased by 1.8% compared to the standard andalusite labeled A60PT, which had a negative effect on increasing the apparent porosity to 16.2% compared to the current product. The lower compressive strength of the sample A60PT0, only CS = 61 MPa (Table 8), was apparently the result of a higher proportion of clay bonds reacting with $Al_2O_3$ additives. The formed mullite phase did not allow sintering, i.e., compaction of the samples during the firing of the sample.

The A60PT4 mixture was designed with an increased content of ceramic bonds at ±17.5%, which is already typical for producing fireclay refractory materials. The next raw materials added were clay with a lower content of $Al_2O_3$ and a plasticizer. The resulting apparent porosity value was lower at 12%, with the highest CS (107 MPa) of the tested samples. The mixture A60PT4 was intended to withstand temperatures of up to 1300 °C and have a heat resistance of $T_{0.5}$ = 1529 °C. The chemical composition with 65% $Al_2O_3$ guaranteed sufficient resistance to melts at this temperature and met the required goals.

*3.2. Experiment II: Corrosion Crucible Test, XRPD, and SEM/EDS Test of Andalusite Refractory Material Labeled A60PT0 and Corrosion Evaluation of A60PT1, A60PT2, and A60PT3*

As mentioned above, the original and new andalusite samples were tested using the crucible corrosion test. $K_2CO_3$ and fly ash after biomass combustion were used for testing. First, A60PT0 was tested, which allowed for a measurement of the infiltration depth by $K_2CO_3$. The intensity of the corrosive attack was evident and is shown in Figure 7A. The tested corrosion agent $K_2CO_3$ had an intensive effect on the depth of infiltration through samples. According to the regulation in [44], the classification of refractory A60PT0 was

defined as corroded "C" and according to the internal regulation in [45], the sample was classified as (D+++) with a large corrosion infiltration. See the smaller photo in Figure 7A.

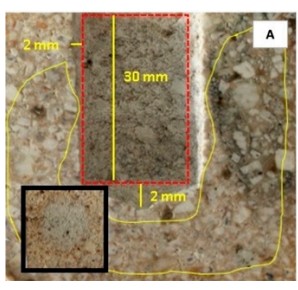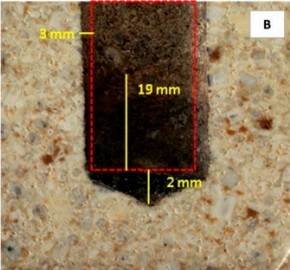

| C | Sample | Corrosion medium | |
|---|---|---|---|
| | | K₂CO₃ | Fly ash |
| | A60PT0 | Class | |
| | ČSN P 15418 | C | A |
| | Internal regulation | D+++ | B+ |
| | Crack | no | no |

**Figure 7.** Andalusite samples after corrosion test. (**A**) Sample A60PT0 with corrosion test parameters of $K_2CO_3$/5 g/5 h/1100 °C; (**B**) sample A60PT0 with corrosion test parameters of fly ash/5 g/5 h/1400 °C; and (**C**) evaluation of crucible corrosion test according to the regulations of [44,45].

Unfortunately, the color of the intensity blends was the same as the color of the sample. The yellow line shows the attack area. The corrosive effect of the fly ash can be seen in Figure 7B. In the case of fly ash, the attack was not as intense as $K_2CO_3$, but there was a slight attack on the edges of the inner part of the sample. An evaluation of refractory materials according to regulations [44,45] is defined as (A) clearly attacked and (B+) a minimally corroded sample. Both samples were without cracks or deformation.

The record of the XRPD phase analysis before and after the corrosion test is presented in Figure 8. The main phase of all tested samples was after the central peak at position 30 and with many peak frequencies from mullite ($3Al_2O_3 \cdot 2SiO_2$) when heated in the temperature range of 1100–1480 °C [47]. The next peaks that were detected were andalusite ($Al_2SiO_5$), $SiO_2$ in cristobalite form, and corundum $Al_2O_3$.

$$3Al_2O_3 \cdot SiO_2 \text{ (andalusite)} \rightarrow 3(3Al_2O_3 \cdot SiO_2) \text{ (mullite)} + SiO_2 \text{ (glass)} \qquad (12)$$

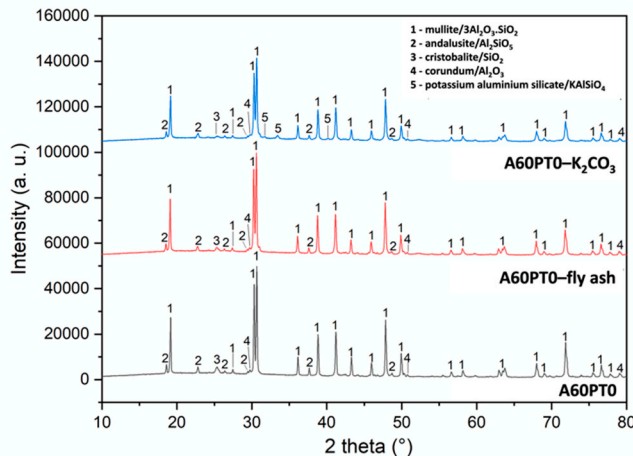

**Figure 8.** X-ray diffractogram of refractory samples A60PT0 in the range of 5–90° 2θ.

The kaliophilite phase, $KAlSiO_4$, was detected after a corrosion attack during 0–5 h at 1100 °C in an andalusite sample tested on $K_2CO_3$. The position of this phase was between 31 and 42 2θ. The mullite peak had a lower visible intensity. This phase was detected by research in the literature [48] within 5 h of a corrosion test at 1000 °C. For example, the leucite phase was recognized after 32 h of attack at the same temperature. The tendency of the kaliophilite peak decreased and a reaction was started between $KAlSiO_4$ and $SiO_2$ to form leucite [25]. A diffractogram of sample A60PT4 after being tested on fly ash showed

a representation of the same phase as the original sample. Volume expansion due to the formation of kaliophilite is responsible for forming cracks in bricks. The reactions in an alumina silica refractory material can be explained: $K_2O + SiO_2 \rightarrow K_2O \cdot SiO_2$ in a glass matrix; $3(K_2O \cdot 2SiO_2) + 3Al_2O_3 \cdot 2SiO_2 \rightarrow 3(K_2O \cdot Al_2O_3 \cdot 2SiO_2) + 2SiO_2$/short time and $K_2O$; and $Al_2O_3 \cdot 2SiO_2 + 2SiO_2 \rightarrow K_2O + Al_2O_3 \cdot 4SiO_2$/long time [25].

The depth of infiltration and the microstructure of the samples were detected by scanning electron microscopy, as shown in Figures 9 and 10. On corroded samples, ten points were detected for the determination of the details of structures and elements (Figure 9). Based on the EDS analysis, a graph was made that indicated the four most frequent elements as Al, Si, Na, and K. Points 1, 2, 8, and 10 were in the area without visible corrosion. On the other hand, points 4, 3, and 6 were in the corroded area. A big gap and changes were visible between points 10 and 4. The elements K and Si decreased, and the alkali compounds slightly increased.

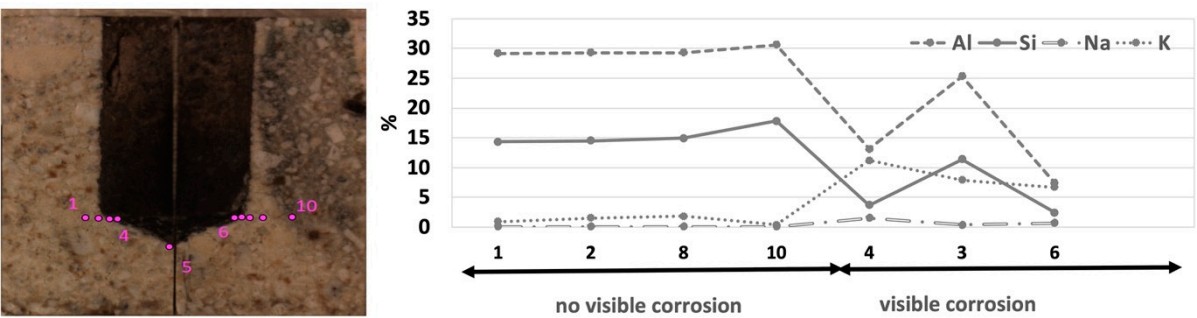

**Figure 9.** Points of measurement for sample A60SPT0 by SEM methods, corresponding to the amount of percentual representation of selected elements obtained from the EDS analysis.

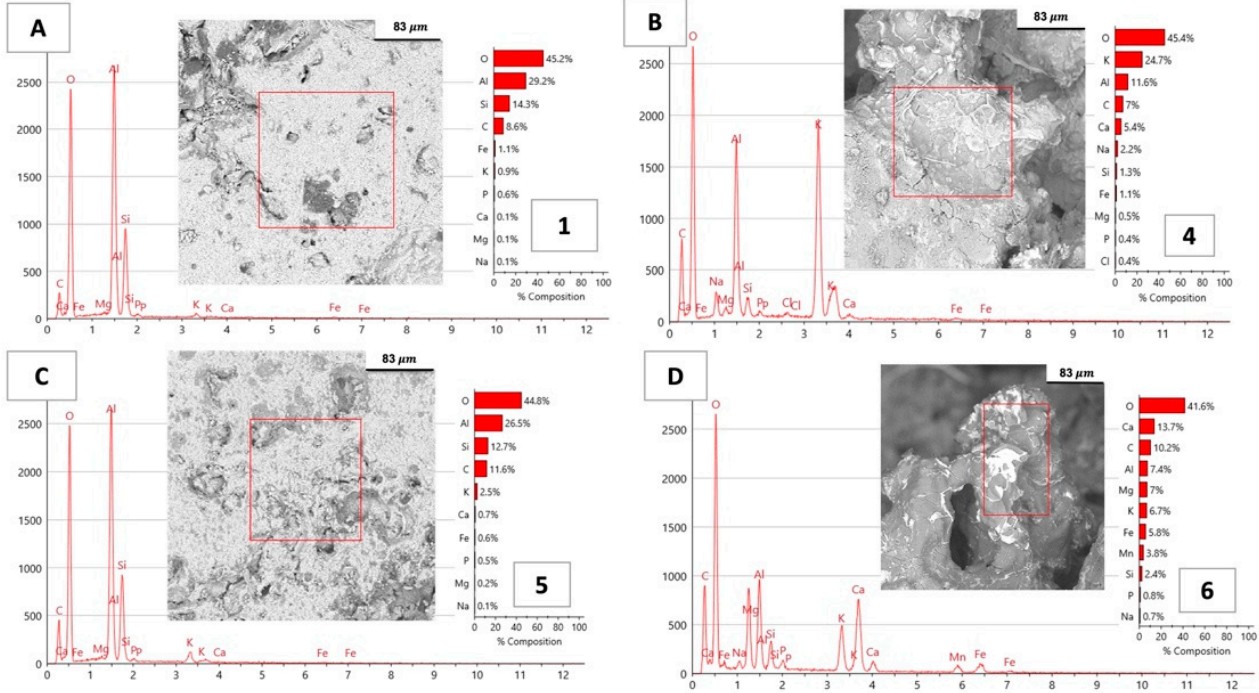

**Figure 10.** SEM/EDS position detected for sample A60PT0 after corrosion test with fly ash at 1400 °C.

Figure 10 shows a different type of structure for sample A60PT0 that is representative of a point of corrosion-tested materials. As observed in this figure, the mullite, glassy, and andalusite phases were formed. Point 1 in Figure 10A indicates the elements Al (29.2%),

Si (14.3%), and O (45.2%) in the mullite phase. This structure had a compacted surface with different asymmetrical grains and a pore. Point 4 in Figure 10B was near the edge of the corrosion attack. It can be assumed to be an andalusite grain containing 24.7% K and 11.6% Al, and a smaller composite grain with 5.4% Ca and 2.2% alkali Na. A heterogeneous structure with many pores and elements such as Al (26.5%) and Si (12.7%) is shown in Figure 10C. Figure 10D shows a visible compacted grain with a complex structure.

The samples A60PT1, A60PT2, and A60PT3 did not fully meet the expected requirements in terms of ceramic properties. However, for the sake of interest, a crucible corrosion test was performed on these samples using $K_2CO_3$, but with a different regime. The results of the corrosion testing of the samples are presented in Figure 11. As can be seen, a higher dose of the corrosion agent resulted in a more intense reaction with the materials and caused infiltration already at a temperature of 1100 °C. Feldspar phases were formed on the surface of sample A60PT2, as shown in Figure 11B,C, by sample A60PT3, which had a stronger corrosion effect. This suggests that the mixtures did not present a high corrosion resistance.

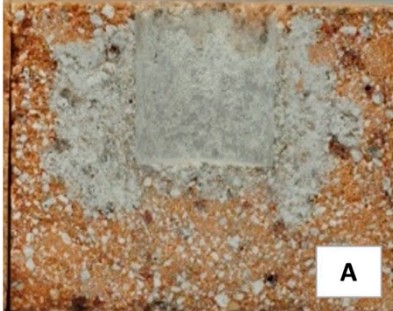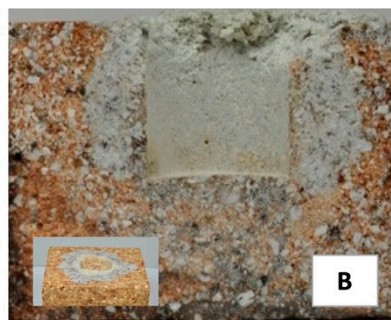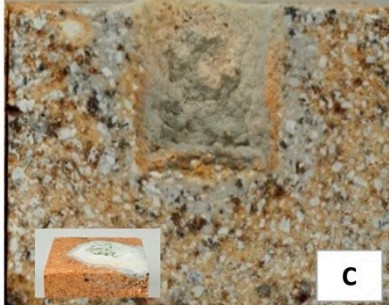

**Figure 11.** Corrosion effect on samples (**A**) A60PT1, (**B**) A60PT2, and (**C**) A60PT3 with corrosion crucible test parameters of $K_2CO_3$/20 g/2 h/1100 °C.

### 3.3. Experiment III: Corrosion Crucible Test, XRPD, and SEM/EDS Test of Andalusite Refractory Material Labeled A60PT4

The new andalusite refractory material labelled A60PT4 contained 60 wt.% $Al_2O_3$, 36 wt.% $SiO_2$, 1 wt.% $Fe_2O_3$, and 1 wt.% $P_2O_5$, with an apparent porosity of up to 13% and use up to 1300 °C. According to ČSN EN ISO 10 081-1 [15], it belongs to group HA 60 of high-aluminum products based on sillimanite with a chemical–ceramic bond. It was assumed that a lower porosity would increase corrosion resistance, even when the question of the wettability of the solid surface of the refractory material with the melt is not solved and the melt has the ability to penetrate the pores.

For the determination of corrosion resistance, a crucible corrosion test was applied. $K_2CO_3$ was used as the corrosion agent in Figure 12A and fly ash was used in Figure 12B. As can be seen from the test results, the refractory material did not melt at a temperature of 1400 °C. The samples did not show expansion or the formation of cracks. However, the infiltration of the $K_2CO_3$ agent was more pronounced than in the case of fly ash alone. The attack of fly ash can be classified as minimal with an assumption to maintain the quality of materials. Subsurface corrosion was visible in the case of $K_2CO_3$ exposure. The evaluation table summarizes the degree of attack resistance according to the above-mentioned methodologies.

According to [44], both andalusite materials had "lightly attacked", slight (C++), and no (B+) corrosion after the internal regulation [45]. An increased temperature (e.g., temperature fluctuations in the furnace aggregates) is a good indicator of changes. In this case, $K_2CO_3$ was an aggressive agent, but the prepared material A60PT4 was resistant to attack. The tested samples had no surface defects or cracks. Depth infiltration was visible on the side of the sample. Compared with the A60PT0 sample, no greater infiltration into the depth of the sample was recorded. In this case, only surface corrosion occurred.

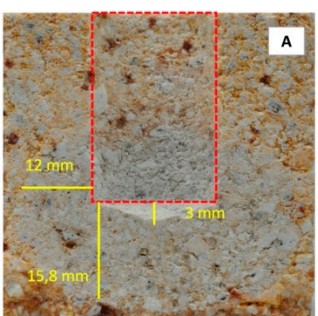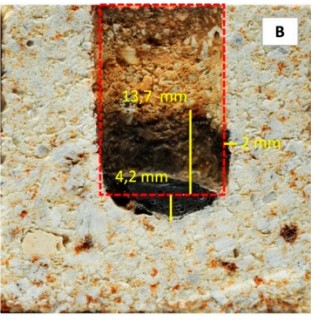

| C | Sample | Corrosion medium | |
|---|---|---|---|
| | | K$_2$CO$_3$ | Fly ash |
| | A60PT4 | Class | |
| | ČSN P 15418 | LA | LA |
| | Internal regulation | C++ | B+ |
| | Crack | no | no |

**Figure 12.** Andalusite samples after corrosion test. (**A**) Sample A60PT4 with corrosion test parameters of K$_2$CO$_3$/5 g/5 h/1100 °C; (**B**) sample A60PT4 with corrosion test parameters of fly ash/5 g/5 h/1400 °C; and (**C**) evaluation of crucible corrosion test regulations according to [44,45].

The flow temperature of fly ash was measured as FT = 1360 °C and the melting temperature was HT = 1361 °C. According to Tuene's number and based on the calculation of the chemical composition of the fly ash, the value of $K_t$ = 0.10 and it was defined as easily fusible fly ash. The calculated value did not correspond to the measured melting temperature. It follows from the above experiments that it is advisable to use multiple test methods for corrosion tests and to apply different test temperatures with many corrosion agents. The testing of refractory materials at high temperatures, for example, at 1400 °C, can be a good indicator of a possible corrosion effect on the refractory material. The use of fly ash as a test agent after biomass combustion is suitable.

In Figure 13, three records of the phase evaluation for the original A60PT4 samples before and after the corrosion test are shown. The main phases of uncorroded A60PT4 were mullite, andalusite/Al$_2$SiO$_5$, and cristobalite/SiO$_2$. These phases were demonstrated for all the samples after the corrosion tests. In the dominant peak, mullite (3Al$_2$O$_3$·2SiO$_2$) was at position 30 2θ. In andalusite corroded by K$_2$CO$_3$, the presence of SiO$_2$ was detected in the form of quartz. The mullite formed from andalusite can be divided into two steps according to the researchers of [30]: below and above 1400 °C. According to the literature [20], the first step consists of the transformation of mullite and a silica-rich liquid phase in a temperature range of 1100–1600 °C, and in the second step, andalusite is transformed into mullite at 1600 °C. A high amount of the oxides CaO and MgO support the speed of forming mullite from andalusite [48]. According to the authors of [49], an amount of 1–2 wt.% dust magnesite added to the biomass as fuel eliminated the attack of the ash sintering and reduced the aggression of ash melt to the refractory material. At the same time, however, the additive must not increase the silica, alkaline, or iron oxide content in the mixture.

The formation of mullite and silica can occur at energetically favorable sites of the andalusite lattice, such as grain boundaries, cleavage planes, and other multidimensional lattice defects [30]. The elemental map from the EDS optical microscopy corresponds to the chemical and phase composition of the andalusite material A60PT4. The elements P, Si, and Al are present in Figure 14A. After testing the K$_2$CO$_3$, the presence of potassium was already detected (Figure 14B). Many types of elements are present in Figure 14C, including Mg and Ca as new main elements.

On the cut sample, A60PT4, the SEM/EDS method was used after the attack of fly ash corrosion, as shown in Figure 15. The individual points 1, 11, 6, and 7, as shown in the figures, were analyzed as no visible corrosion area, and points 12 to 15 were analyzed in the corrosion area. The curves correspond to the % representation of elements from the measured area, which were visually evaluated as non-corroded parts/corroded parts.

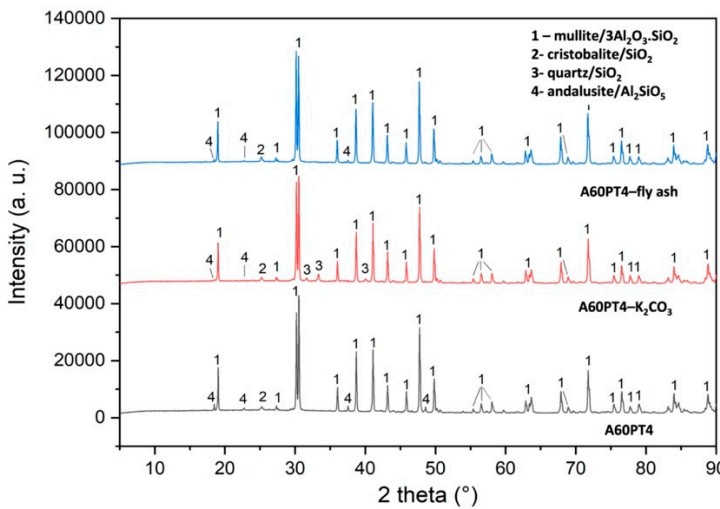

**Figure 13.** X-ray diffractogram of refractory sample A60PT4 in the range of 5–90° 2θ.

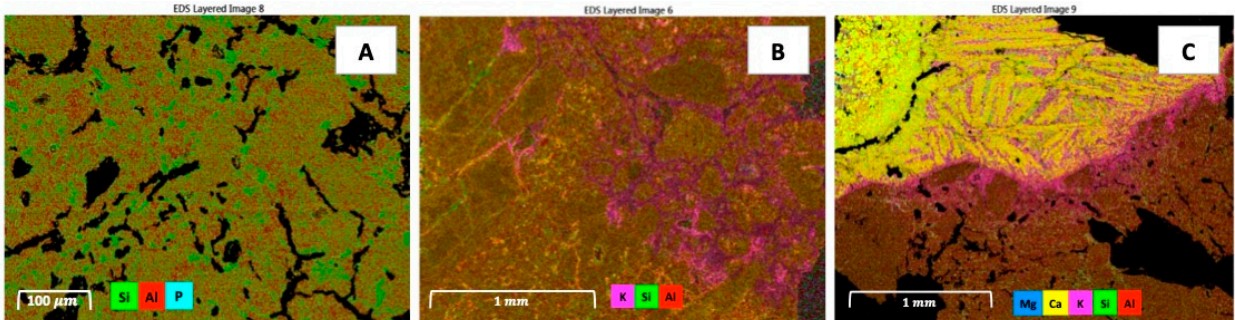

**Figure 14.** EDS analysis of andalusite refractory A60PT4 before and after corrosion test; (**A**) original A60PT4; (**B**) A60PT4 after corrosion by $K_2CO_3$; and (**C**) A60PT4 after corrosion by fly ash.

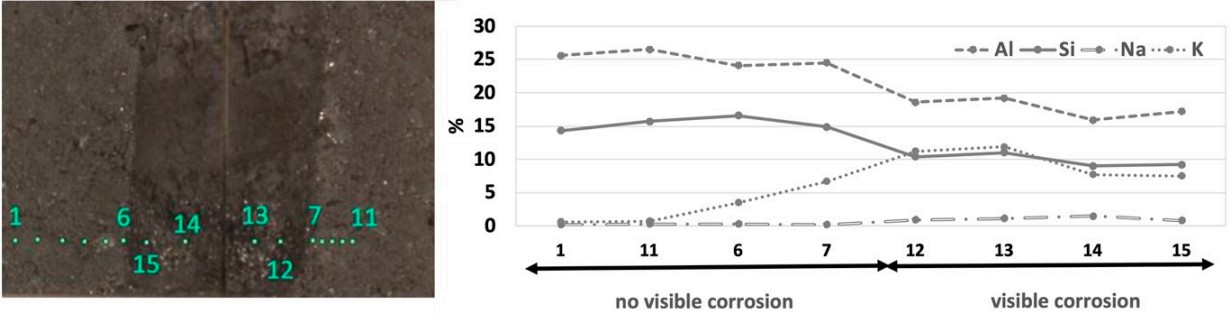

**Figure 15.** Points of measurement for sample A60PT4 by the SEM method, corresponding to the amount of percentual representation of selected elements obtained from the EDS analysis.

Figures 15 and 16 correspond to each other. The detected area of point 1, shown in Figures 15 and 16A, contained 25.6% Al, 14.3% Si, and 3.6% C, with the other elements making up <1% of the content. Position 6 was visually in the same place.

Position 6 can be named as the second area of the beginning of the corrosion effect on the "edge", as shown in Figures 15 and 16C. The amount of the Si content increased slightly with a value of 16.8%, and a trend was observed for potassium at 3.5%. The amount of alkalis was around 5.9%. Point 13 in the corroded area (Figures 15 and 16B) contained around 11% K, 18% Al, and 11% Si.

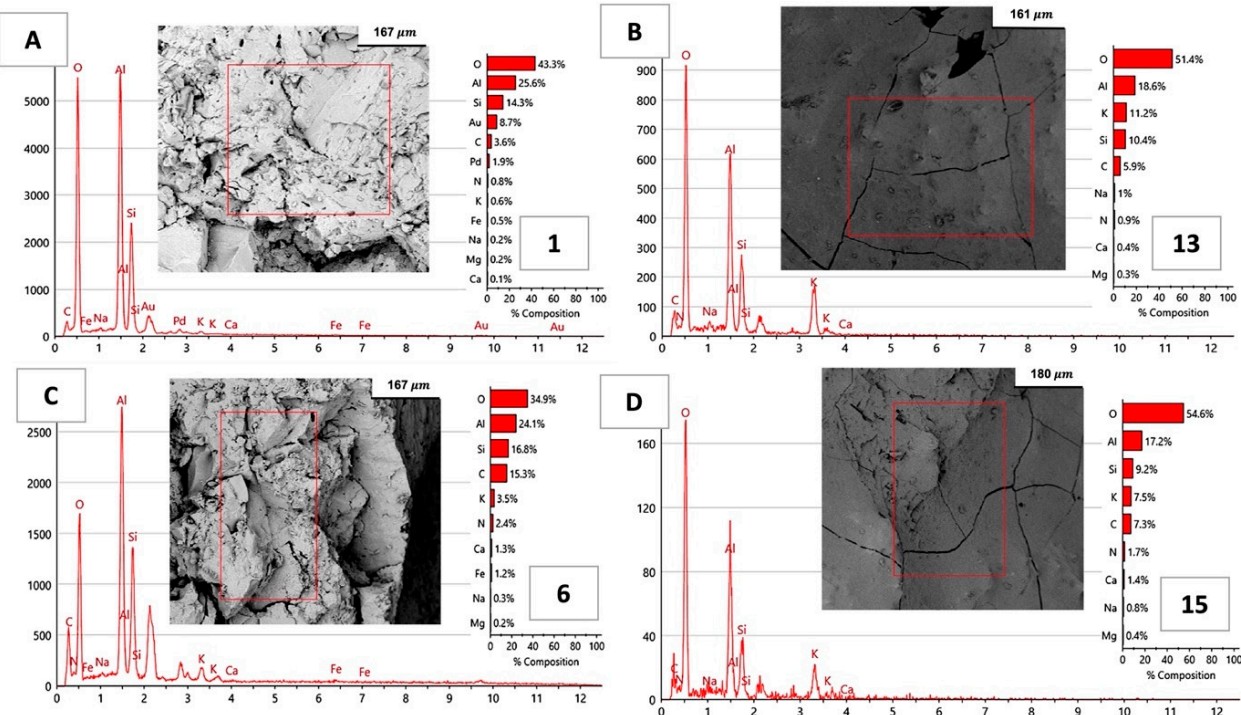

**Figure 16.** SEM/EDS analysis of andalusite refractory material labeled A60PT4 after corrosion test with fly ash at 1400 °C.

Points 12 and 13 were were very similar with an elementary amount. For the last point, 15 (Figures 15 and 16D), the potassium content decreased to a value of 7.2%, and the same trends were observed for the element Al, with 17.2%. The results of the scanning electron microscope (Figure 16A,C) corresponded with points 1 and 6; the structure of the samples was not compact and hard to identify in shape. On the contrary, points 13 and 15 (Figure 16B,D) stood out with a compact sintered surface with visible microcracks and pores. This was caused by a fly ash coating that filled the microscopically visible pores and formed a layer on the surface of the A60SPT4 refractory material.

## 4. Conclusions

During biomass combustion, it is very problematic to maintain the quality of biomass and the declared temperature regime in the furnace. The final products after burning biomass are solid products such as fly ash and slag with a heterogeneous composition. These can melt or form deposits and degrade materials such as steel and refractory materials in the device. These by-products influence the corrosion of the working lining of furnace units. Alkaline corrosion is a big problem with aluminum silica refractory materials. To assess the degree of corrosion and the influence of degradation, the following must be known: (a) the properties of products created during the combustion of biomass and (b) the operating conditions of furnace aggregates.

Furnace aggregates for burning biomass have operating temperatures of up to 1200 °C, at which temperature a melt may not form. However, it often happens that there is a short-term exceeding of the operating temperature up to 1400 °C caused by fluctuations in the calorific value of the biofuel. For this reason, it is recommended to use refractory materials with a higher content of $Al_2O_3$ (above 60%). The modification of the mixture composition was mainly aimed at achieving a higher density and reducing the porosity from the original value of 15% to 11%, which would ensure less infiltration of unwanted oxides.

Other parameters of sample A60PT4 were a volumetric weight of 2679 kg/m$^3$ and a compressive strength of CS = 107 MPa. The newly prepared material A60PT4 contained mainly oxides, 32% $SiO_2$, and 65% $Al_2O_3$ and was resistant to corrosion by $K_2CO_3$ and

the tested fly ash obtained from biomass combustion. According to the regulations and internal standards, this is a new material adapted to resist the environment of biomass combustion. For sample A60PT4, new phases were not detected after the reaction of fly ash and $K_2CO_3$, as determined by an X-ray analysis.

**Author Contributions:** Conceptualization, H.O.; methodology, M.T. and J.V.; investigation, M.T., J.B. and P.M.; writing—original draft preparation, H.O. and J.V.; writing—review and editing, H.O. and J.V.; visualization, H.O.; supervision, M.V.; project administration, M.V.; funding acquisition, M.V. All authors have read and agreed to the published version of the manuscript.

**Funding:** This research was funded by the Ministry of Education, Youth and Sports of the Czech Republic via the Research and Development of Multifunctional Materials for Sustainable, grant number SP2023/034, and the Research on the Management of Waste, Materials and Other Products of Metallurgy and Related Sectors, grant number CZ.02.1.01/0.0/0.0/17_049/0008426.

**Data Availability Statement:** The data presented in this study are available from the corresponding author upon request.

**Conflicts of Interest:** The authors declare no conflict of interest.

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
