# Peer review of "The Corrosion Effect of Fly Ash from Biomass Combustion on Andalusite Refractory Materials"

_minerals, doi:10.3390/min13030357_

Round 1
Reviewer 1 Report
The authors characterized in detail the studied refractory materials in the energy industry, taking into account the conditions of biomass combustion (temperature, NOx, COx, etc.). They characterized the tested bio-ashes, and used relevant analytical methods and equipment. To assess the corrosion resistance, they used a static crucible test and used K2CO3 and selected bioash as corrosion media. ČSN P CEN/TS 15 and Internal Regulation for Corrosion Testing of Refractory were used to evaluate corrosion tests; P-D Refractories CZ. By modifying the formula of the high-aluminum refractory material A60PT0, they monitored corrosion resistance. The obtained results were qualitatively and clearly documented and evaluated. I have the following comments on the submitted work:
- the sequence of chemical reactions (4) to (11) should be assessed according to the thermodynamic analysis of reactions (e.g. HSC),
- the effect of the porosity of refractory materials on the course of corrosion can also be influenced by changing the wettability of the surface with the given melt,
- in the description of the corrosion test methodology (236 - 259) it is necessary to indicate the temperature of the test, the time of exposure to the temperature, the amount of corrosion medium, or temperature characteristics of the furnace (°C/min),
- table 4 lacks an explanation for C = CaO,
- line 208 and table 4 what is "Tueneho number"
Fig. 12 - what was the melting temperature of the ash (line 409-411), the melting temperature of K2CO3 is 891°C / corrosion test 1100°C / degree of superheating of the melt is approx. 200°C, how does it work in the case of bioash?
Author Response
Reviewer 1
Dear reviewer, thank you very much for your comment, spent time and positive reaction on the topic. All English language recommendations have been corrected as recommended. English language was editing to quality for acceptable to print.
Thank you for commnet and and below are the answers.
The authors characterized in detail the studied refractory materials in the energy industry, taking into account the conditions of biomass combustion (temperature, NOx, COx, etc.). They characterized the tested bio-ashes, and used relevant analytical methods and equipment. To assess the corrosion resistance, they used a static crucible test and used K2CO3 and selected bioash as corrosion media. ČSN P CEN/TS 15 and Internal Regulation for Corrosion Testing of Refractory were used to evaluate corrosion tests; P-D Refractories CZ. By modifying the formula of the high-aluminum refractory material A60PT0, they monitored corrosion resistance. The obtained results were qualitatively and clearly documented and evaluated. I have the following comments on the submitted work:
- the sequence of chemical reactions (4) to (11) should be assessed according to the thermodynamic analysis of reactions (e.g. HSC)/Thanks for the reminders. In the article reaction, 4-11 are presented only from cited sources. Equals are documented in the theoretical part and were not the subject of further closer examination.
- the effect of the porosity of refractory materials on the course of corrosion can also be influenced by changing the wettability of the surface with the given melt/this idea was added to the part 3.3
- in the description of the corrosion test methodology (236 - 259) it is necessary to indicate the temperature of the test, the time of exposure to the temperature, the amount of corrosion medium, or temperature characteristics of the furnace (°C/min)/corrected
- table 4 lacks an explanation for C = CaO/ corrected
- line 208 and table 4 what is "Tueneho number" / In the case of ash, it is also possible to determine the fusibility of the ash according to the coefficient of the so-called „Tuene number“, which is the ratio of acidic and basic oxides. The autor use the inspiration in ling below
https://docplayer.cz/4527821-Charakteristiky-paliv.html
Fig. 12 - what was the melting temperature of the ash (line 409-411), the melting temperature of K2CO3 is 891°C / corrosion test 1100°C / degree of superheating of the melt is approx. 200°C, how does it work in the case of bioash? /Flow temperature of fly ash was FT = 1360 °C and the melting temperature was HT = 1361°C. For determinig of temperature was used method by according to the STN ISO 540 on a thermal microscope. According to the Ash fusibility index (AFI), if the value AFI >1342°C means that the tendency of ash to slagging/fouling is low. According to Tuene's number based on the calculation of the chemical composition of the fly ash, the value of Kt = 0.10 and is defined as easily fusible ash. The calculated value does not correspond to the measured melting temperature. It follows from the above experiments that it is advisable to use multiple test methods for corrosion tests and to use different test temperatures with different corrosion agents. Higher test temperatures can be a good indicator of a possible corrosion effect on the refractory material being tested, in the event that undesirable temperature fluctuations occur during combustion. The use of ash after biomass combustion as a test agent appears to be useful, including the possibility of calculating ash indices.
Reviewer 2 Report
This manuscript investigated the corrosion of the fly ash on andalusite refractories, but some details need to be improved before published.
1.Some terms in this manuscript are not standardized and some expressions are colloquial. The full text needs to be carefully checked and revised
Here are some examples of the confusing expressions:
Page 5 Line 164 “(AP÷15 %)”
Page 8 Line 254 “(not sure what you mean 254 here. Maybe need to be more specific)”
Page 15 Line 481 “higher content”
2. Evaluation of corrosion tests
The measurement accuracy of penetration depth is not reliable enough. The use of optical microscope or scanning electron microscope can provide higher accuracy than visual evaluation.
3. The purity or chemical composition of the erosion medium should be provided.
4.Figure 16 The comparison of element content on uneven surface lack of reliability.
5. The language and grammar of this text should be checked and revised with the help of a native English speaker.
Author Response
Reviewer 2
Dear reviewer, thank you very much for your comment, spent time and positive reaction on the topic. All English language recommendations have been corrected as recommended. English language was editing to quality for acceptable to print.
Rewiever 2
This manuscript investigated the corrosion of the fly ash on andalusite refractories, but some details need to be improved before published.
1.Some terms in this manuscript are not standardized and some expressions are colloquial. The full text needs to be carefully checked and revised
Here are some examples of the confusing expressions:
Page 5 Line 164 “(AP÷15 %)” /corrected
Andalusite refractory material has high compressive strength characteristics, high load capacity in heat, and dimensional stability. However, it has a lower density, i.e., a higher apparent porosity of 15% associated with lower corrosion resistance and easier penetration into refractory material.
Page 8 Line 254 “(not sure what you mean 254 here. Maybe need to be more specific)”/corrected
After cooling the crucible was vertically cut, and the penetration of refractory material was measured. The affected area is otherwise evaluated. The advantage of the test is its simplicity and quick feedback on the result. Evaluation of corrosion tests can often be subjective and often depend on visual evaluation and operator or research experience. The study was evaluated according to a norm [44] with individual classes described in Table 5 and an internal evaluation in Table 6.
Page 15 Line 481 “higher content”/corrected
- Evaluation of corrosion tests
The measurement accuracy of penetration depth is not reliable enough. The use of optical microscope or scanning electron microscope can provide higher accuracy than visual evaluation.
Thank you for reminder. I certainly agree. However, the experiment was based on the fastest corrosion evaluation. In practice the companies do not often have these devices. Therefore, using a prescription or self-assessment is often the most effective. The author therefore relies on the simple’s possible evaluation with adequate results.
- The purity or chemical composition of the erosion medium should be provided./added to the text now page 7
2.3. Characterization of K2CO3
As the second corrosive medium was used anhydrous potassium carbonate. This product is from the company Penta s.r.o., Czech Republic. Information of chemica and physical properties is declared in safety data sheet: colour – white, state – solid, melting point 891°C and content in wt.% > 99.
4.Figure 16 The comparison of element content on uneven surface lack of reliability.
Thank you for reminder. I agree with you. I didn’t made measurement personally, but in future will be will be proceed more thoroughly. Point of view of granulometry refractory are more complicated and the researcher which will be made measurement have to use more step for analysis.
Round 2
Reviewer 2 Report
The comments of reviewer have been properly replied. The manuscript can be accepted now.